# Investigation of Material Removal Distributions and Surface Morphology Evolution in Non-Contact Ultrasonic Abrasive Machining (NUAM) of BK7 Optical Glasses

**DOI:** 10.3390/mi13122188

**Published:** 2022-12-10

**Authors:** Zongfu Guo, Xichun Luo, Xiaoping Hu, Tan Jin

**Affiliations:** 1School of Mechanical Engineering, Hangzhou Dianzi University, Hangzhou 310018, China; 2Centre for Precision Manufacturing, DMEM, University of Strathclyde, Glasgow G1 1XJ, UK; 3College of Mechanical and Vehicle Engineering, Hunan University, Changsha 410082, China

**Keywords:** non-contact ultrasonic abrasive machining (NUAM), optical polishing, material removal distributions, surface morphology

## Abstract

A non-contact ultrasonic abrasive machining approach provides a potential solution to overcome the challenges of machining efficiency in the high-precision polishing of optical components. Accurately modeling the material removal distribution (removal function (RF)) and surface morphology is very important in establishing this new computer-controlled deterministic polishing technique. However, it is a challenging task due to the absence of an in-depth understanding of the evolution mechanism of the material removal distribution and the knowledge of the evolution law of the microscopic surface morphology under the complex action of ultrasonic polishing while submerged in liquid. In this study, the formation of the RF and the surface morphology were modeled by investigating the cavitation density distribution and conducting experiments. The research results showed that the material removal caused by cavitation bubble explosions was uniformly distributed across the entire working surface and had a 0.25 mm edge influence range. The flow scour removal was mainly concentrated in the high-velocity flow zone around the machining area. The roughness of the machined surface increased linearly with an increase in the amplitude and gap. Increasing the particle concentration significantly improved the material removal rate, and the generated surface exhibited better removal uniformity and lower surface roughness.

## 1. Introduction

Brittle materials, such as Si, SiC, and BK7 optical glass, have been widely used for high-end optical applications for precision detection and navigation. CNC grinding and polishing are usually used to machine them into the desired shape. Although the form accuracy of the ground surfaces can reach the scale of 1 μm/100 mm, the subsurface damage is in the scale of tens of microns to hundreds of microns, which often results in a long polishing time to remove the damaged layer [1,2]. A highly efficient polishing process is therefore required to reduce the machining time for these precision optics. For flat and spherical optics, ring polishers and swing-arm machines are often used to achieve fast polishing [3,4,5]. For aspheric optics, a small-tool polishing technique [6] is used to correct the sub-aperture surface error and remove the subsurface damage of the grinding surfaces. Bonnet polishing [7,8], magnetorheological finishing (MRF) [9,10], ion beam figuring (IBF) [11,12], and fluid jet polishing (FJP) [13,14] fall into this category, where a sub-nanometer surface finish, a nanometer to tens of nanometers surface form accuracy, and a nanometer-level or zero subsurface damage can be obtained. However, machining efficiency remains a significant challenge in these ultra-precision polishing technologies.

To improve the polishing efficiency, many researchers combine ultrasonic processing technology with the existing polishing technology by applying ultrasonic vibrations to the polishing tool, workpiece, or polishing fluid. For example, Zhang studied a rotary polishing tool head applied vertically to the surface of BK7 optical glass with the aid of axial ultrasonic vibration (UVAP). The experimental results show that this method can improve both the material removal rate (MRR) and surface quality [15,16]. Suzuki developed a two-axis ultrasonic vibration polishing tool to obtain more complex polishing trajectories. Experiments showed that applying ultrasonic vibration to a polishing tool allows for obtaining a surface roughness Rz of 8 nm [17]. In abrasive waterjet machining, Lv adhered a workpiece to an ultrasonic horn and the workpiece was ultrasonically vibrated by the horn. The experimental results show that the introduction of ultrasound can improve the material removal efficiency by 82% [18,19]. Beaucamp studied the method of applying ultrasonic motion inside a jet nozzle to improve the removal rate of fluid jet polishing and the material removal rate was increased by 380%. The research results of ultrasonic-assisted jet polishing show that the jet fluctuation caused by ultrasonic vibration is the main factor that improves the MRR. As the ultrasonic cavitation zone is far away from the workpiece surface, the cavitation bubbles disappear before reaching the workpiece surface, and thus, the microjets caused by the explosion of the cavitation bubbles did not affect the material removal [20,21].

In addition, Ichida et al. proposed a new non-contact ultrasonic abrasive machining (NUAM) method to machine aluminum alloy (JIS-2014). A schematic of NUAM is shown in Figure 1, where an ultrasonic horn is attached to the ultrasonic transducer and used as a processing tool. There is a gap between the working surface of the ultrasonic horn and the surface of the workpiece that is filled with polishing liquid. The ultrasonic vibration acts on the surface of the workpiece through the polishing liquid to achieve material removal. Three types of material removal modes were found in Ichida’s study. They are direct impact removal of microjets produced by cavitation bubble explosion, impact removal by microjet driven abrasive grains, and scratch removal from abrasive grains driven by fluid drag [22]. However, there is no relevant research on the material removal distribution and the evolution of the surface micromorphology of brittle materials, such as optical glass, under this new processing method.

Among the aforementioned ultrasonic-assisted polishing methods, NUAM technology requires the most simplified device and has the least restrictions on the processing object. It can also avoid problems such as abrasive particle agglomeration and tool wear better than other discrete abrasive polishing methods. This study aimed to investigate the feasibility of further developing NUAM technology into a new high-efficiency polishing method for optical materials. Two key challenges to achieving ultra-precision polishing by NUAM are efficient deterministic material removal and smooth machined surfaces. Therefore, this study carried out an experimental study on material removal modes, the relationship between the proportion of different removal modes and the MRR and surface quality, and the relationship between the material removal distribution and surface quality and the associated process parameters in the NUAM of BK7, which is a typical brittle material used in the optics industry.

This paper is arranged as follows: Section 2 gives a theoretical analysis of material removal distribution, Section 3 describes the conducted experiments and analysis results, and Section 4 presents the material removal distribution and microscopic surface formation modeling results.

## 2. Analysis of Material Removal Distribution under NUAM

In a polishing process, the polishing liquid is mainly a suspension that is formed by mixing hard abrasive particles and deionized water. Under the action of ultrasonic vibration, the air dissolved in the liquid water will separate to form bubbles, which become cavitation bubbles. Cavitation bubbles grow, shrink, collapse, or redissolve as the pressure fluctuates [23,24]. Exploding cavitation bubbles will create high-velocity microjets that enable material removal. This section analyzes the material removal distribution caused by cavitation and polishing fluid scouring via theoretical analysis and computer simulation.

### 2.1. Theory of Cavitation

The material removal rate and material removal distribution caused by cavitation are closely related to the distribution density, collapse strength, and collapse position of ultrasonic cavitation bubbles. Schnerr and Sauer presented a cavitation model to predict the cavitation distribution based on a combination of the VOF (volume of fluid) technique with a model predicting the growth and collapse process of bubbles. The void fraction (α), defined as the volume of vapor divided by the cell volume, can be expressed as Equation (1) [25]:(1)$$\alpha =\frac{n_{0}\cdot \frac{4}{3}\pi R^{3}}{1+n_{0}\cdot \frac{4}{3}\pi R^{3}}$$
where *n*_0_ is the number of bubbles in a unit volume of fluid and *R* is the average cavitation bubble radius. The bubble radius changes with the fluid pressure and can be expressed by the Rayleigh relation, as shown in Equation (2) [25]:(2)$$R=\sqrt{\frac{2}{3}\frac{p\left(R\right)-p\infty }{\rho_{l}}}$$
where *p*(*R*) is the pressure in the liquid at the bubble boundary, *p*∞ is the pressure in the liquid at a large distance from the bubble, and *ρ_l_* is the fluid density. Cavitation will occur when the liquid additional negative pressure reaches the cavitation threshold. According to the condition of the pressure balance between the inside and outside of the cavitation bubble, Equation (3) [26] was proposed by Neppiras for solving the cavitation threshold:(3)PB=P0−PV+23[(2σR0)33(P0−PV+2σR0)]12
where *P*_0_ and *P*_V_ are the hydrostatic pressure and saturated vapor pressure in the bubble, respectively; σ is the liquid surface tension coefficient; and *R*_0_ is the initial radius of a cavitation bubble. In the research, a high-speed camera is usually used to observe cavitation bubbles [20,27,28]. The diameter of cavitation bubbles observed in the experiment is generally at the scale of several microns to tens of microns. When ultrasonic vibration acts directly on the water, the cavitation bubble diameter is 10 μm, the saturated vapor pressure is *P_V_* = 2.34 × 10^3^ Pa, and *σ* = 0.0728 N/m at 20 °C, the cavitation threshold *P*_B_ is 9.968 × 10^4^ Pa.

According to Luther’s work, the average lifetime of cavitation bubbles is equal to 200 acoustic pressure cycles [28]. When the ultrasonic vibration frequency is 26 kHz, the observation results of Beaucamp’s work were also very close to this conclusion [20]. The ultrasonic vibration frequency used in this study was 28 kHz, and thus, the lifetime of the cavitation bubbles was considered to be 7 ms. For the polishing fluid suspension, the contained microparticles will increase the cavitation erosion by increasing the number of cavities in the suspension [29,30].

Plesset and Chapman calculated that the microjet velocity is about 130 m/s when a bubble collapses while attached to a wall and about 170 m/s when a bubble collapses near the wall [31]. The pit depth is determined by both the microjet velocity and microjet diameter and increases with their increase. The pit diameter (*d_p_*) is mainly related to the microjet diameter (*dj*), where *d_p_*/*d_j_* ≈ 0.95–1.2, while the pit’s diameter-to-depth ratio is mainly negatively correlated with the microjet velocity [32].

### 2.2. Simulation of Cavitation

Using the computer fluid simulation analysis software Fluent combined with the dynamic mesh technology to simulate the drive of the ultrasonic tool head to the polishing liquid, the pressure and velocity distributions exerted by the ultrasonic vibration on the surface of the workpiece through the polishing liquid were analyzed, along with the distribution of the generated cavitation clouds. Deionized water and hard abrasive particles, such as cerium oxide and alumina, are usually used in polishing, and the suspension formed by mixing is used as a polishing liquid. The mass concentration of abrasive particles is generally 5–40 g/L. At this concentration, the flow characteristics of the suspension and water are very similar, and thus, water is used as the fluid analysis object. Figure 2a shows the finite element model.

Without affecting the purpose of the study, the following simplified conditions were used: the effect of particles on the cavitation was ignored, the temperature was kept constant, water was used as the polishing liquid, and a 2D axisymmetric model was used. The detailed simulation parameter settings are shown in Table 1.

The simulation results showed that the liquid velocity caused by ultrasonic vibration reached the maximum at the edge of the gap, as shown in Figure 2b. The pressure in the gap changed periodically with the ultrasonic vibration and was roughly distributed in a Gaussian shape. The maximum velocity at the edge of the gap and the maximum pressure inside the gap decreased exponentially with the increase in the gap and showed a rapidly decreasing trend with the decrease in the amplitude, as shown in Figure 3.

The variation in cavitation clouds generated by ultrasonic vibration with time was represented by the variation in vapor fraction shown in Figure 4. It can be seen from this figure that under the high-frequency impact of the polishing liquid, the cavitation zone first appeared near the working surface of the horn, the cavitation bubbles overflowed at the edge of the gap in the radial direction, and the overflowing cavitation bubbles moved away from the workpiece surface. When moving away from the surface of the workpiece, the overflowing cavitation bubbles dissolved or disappeared after some time and eventually reached equilibrium.

Figure 5 shows the variation trend of cavitation intensity and cavitation distribution with respect to the gap and amplitude. It can be seen from this figure that increasing the amplitude significantly improved the cavitation rate, while increasing the gap had little effect on the cavitation rate. When the gap was large, if the cavitation rate was not enough, the cavitation cloud gathered near the working face and could not fully exert the erosion processing on the workpiece.

### 2.3. Removal Theory Using Fluid Scouring

In NUAM, the polishing fluid is driven by an ultrasonic vibration tool to perform high-frequency scouring on the surface of the workpiece. This fluid scouring removal method is very similar to the fluid jet polishing method. Based on the erosion model for FJP and ultrasonic FJP [34], the removal volume (*V*) of a single abrasive driven by ultrasonic vibration can be described using Equation (4) [21]:(4)$$V=k\frac{1}{4}m_{p}v_{vib,r}^{2}{\left(\frac{1}{4}m_{p}v_{vib,z}^{2}\right)}^{\frac{2\left(1-b\right)}{3}}$$
where *v_vib,r_* and *v_vib,z_* respectively represent the average values of the vibration velocity along the radius and perpendicular to the surface of the workpiece per unit time, *m_p_* is the abrasive particle mass, *k* is a material-dependent coefficient that accounts for plastic flow pressure and material springback, and *b* (0.5 ≤ b ≤1) is a material-dependent exponent related to the cross-sectional area of the abrasive indentation.

### 2.4. Prediction of the Material Removal Distribution

From the distribution of cavitation clouds in the gap, it can be predicted that the material removal caused by cavitation blasting is uniformly distributed in the gap. Due to the cavitation overflow at the edge of the gap, this processing method will have obvious edge effects.

In the fluid jet polishing process, the maximum impact velocity of the polishing liquid on the workpiece surface is usually about 44 m/s (jet pressure = 1 MPa), and the removal depth is at the scale of 1 nm/s [13] and increases exponentially with the increase in the jet pressure [35]. According to the maximum speed and pressure shown in Figure 3, when the amplitude was less than 30 μm and the gap was greater than 0.15 mm, the material removal rate caused by scouring was very low. Through the above Equation (4), it is found that the erosion removal ability of a single abrasive particle mainly depends on its tangential velocity on the workpiece surface. If the abrasive particles are evenly distributed in the polishing solution, it can be inferred that the material erosion removal distribution is consistent with the change curve of the tangential velocity. The maximum velocity according to Figure 2b is located at the edge of the gap, and thus, the distribution of material removal caused by scouring is also concentrated at the edge.

Based on the above analysis, the tool’s removal function (*RF*) of this method can be described as follows:(5)$$RF=S_{surface}+S_{edge}+S_{scour}$$
where *S_surface_* is the shape of the tool’s working surface, *S_edge_* is the edge removal distribution, and *S_scour_* is the removal distribution due to fluid scouring.

## 3. Experimental Research and Results Analysis

### 3.1. Experimental Preparation

Figure 6 shows the experimental setup. The ultrasonic frequency was 28 kHz. The ultrasonic horn worked as a tool and three kinds of tool shapes were used (a plane with a diameter of 8 mm, a plane with microgrooves, and a plane with a protruding five-pointed star). The amplitude of the working face of the horn could be adjusted by adjusting the output power. The workpiece was a flat mirror with a diameter of 50 mm and the material was BK7. The surface roughness (*Sa*) of the workpiece before processing was 0.5 nm. The workpiece was submerged in the polishing fluid to a depth of 20 mm. The fluid was filled with CeO_2_ abrasive particles, where the average diameter of the particles was 1 micron. The detailed experimental conditions setting are shown in Table 2.

### 3.2. Material Removal Distribution

#### 3.2.1. Under the Action of the Plane Tool

The horn with the flat working surface was used to carry out a fixed-point processing experiment with different process parameters. Figure 7 shows the removal pit shapes, which were measured using a laser interferometer (Zygo GPI).

The measurement results clearly showed that the distribution of the removed material was like a tool head shape (round). Figure 7e shows the lens’s overall surface shape error distribution after processing. By analyzing the shape changes in the bottom surface of the removal pits and the pit’s position on the lens, it was found that the initial surface shape of the lens was retained. This showed that the changing trend in the overall surface shape at the pits was consistent with the changing trend at the bottom of the pits, which meant that the material removal in the tool processing surface was uniformly distributed. There was a groove on the edge of the removal pit, which can be seen clearly in Figure 7b,d, and the groove depth (*d_J_*) increased with the increase in removal depth (*d_U_*), as seen by comparing the cross-sectional profile. The ratio of *d_J_* and d_U_ changed under different process parameters. For example, the ratio of d_J_ and *d_U_* in Figure 7b was clearly larger than in Figure 7d. The width of the groove was about 0.5 mm. The center of the removal area was slightly raised, which can be seen clearly in Figure 7c. The diameters of removal pits under different process parameters were basically the same at about 8.5 mm, which was 0.5 mm larger than the tool diameter.

Based on the simulation results of the speed and pressure distribution in the gap (Figure 2), the fluid speed increased sharply in the machine edge and the max value decreased as the gap increased; therefore, we can conclude that the groove was generated by the fluid scouring. The material removal in the processing area was basically evenly distributed, and it can be concluded that the polishing liquid scouring removal in the processing area played a secondary role and the cavitation jet scouring removal played a major role. Therefore, the ratio of the pit removal depth *d_J_* to the average removal depth d_U_ reflects the ratio of jet erosion removal to ultrasonic cavitation microjet removal in the vicinity. The edge effect was around 0.25 mm, which indicated that the overflowing cavitation cloud did not produce significant material removal. This may have been because the pressure fluctuation outside the action zone was small, most of the cavitation bubbles redissolved and disappeared, or the distance away from the workpiece surface under the action of the drag force exceeded the action range of the blasting jet.

#### 3.2.2. Under the Action of the Tool with a Complex Structure

Figure 8 shows the material removal distribution formed by different geometric structures on the working surface of ultrasonic vibration tools. It can be seen from Figure 8a that interference formed when the spacing of the geometric structures was less than 0.5 mm. It did not matter whether the structure was inside or at the edge of the ultrasonic vibrating surface. This further indicated that the material removal caused by fluid scouring was small. It can be seen from Figure 8b that the corresponding removal distribution at the apex of the five-pointed star became an excessive arc, where the radius of the arc was about 0.25 mm. It can be known from the above experimental results that this processing method produced an edge effect of 0.25 mm. Therefore, if the geometric structure of the tool’s working surface is larger than 0.5 mm, the structure can be copied to the workpiece surface using this method.

#### 3.2.3. Interaction Analysis of the Removal Pits

Figure 9 shows the material removal distribution under different coverage ratios formed by adjusting the center distance during two machining operations using an ultrasonic vibration tool with a diameter of 8 mm. It can be seen from the figure that there was almost no mutual interference between the two processing operations; therefore, peaks or valleys of different scales can be obtained by adjusting the center distance during the two processing operations, and this feature also satisfies the deterministic polishing process. This means that in the future, the processing of special-shaped complex parts can be realized, such as commonly used etalons and array structures, by combining path planning, tool head geometry design, and deterministic polishing control technology.

### 3.3. Removal Depth and Surface Morphology

The morphology of the removal pit’s bottom was observed using a white light interferometer (Zygo Newview, Connecticut, United States). The plane ultrasonic horn was used to carry out the experiment to investigate the influence of the process parameters.

Figure 10a shows that the removal depth increased exponentially and the surface roughness increased linearly with the increase in amplitude. It can be known from the above analysis that as the amplitude increased, the cavitation density and cavitation intensity also increased, which meant that more and stronger cavitation bubbles participated in the material removal, increasing the removal rate. Figure 10b shows that the surface morphology exhibited more obvious discreteness and randomness compared with the surface formed by small tool head polishing or jet polishing [36]. As the amplitude increased from 20 μm to 40 μm, the removal uniformity became significantly worse and it appears that the red area representing the convex peaks on the figure became smaller and more dispersed. This was because the blasting energy of the cavitation bubbles increased, thereby increasing the removal depth. Furthermore, after the amplitude increased to 50 μm, the red area blocks on the figure became significantly larger. This was because the peaks formed on the surface became slenderer as the amplitude increased and the polishing fluid flow velocity in the gap increased, resulting in the removal of the microstructure peaks.

The above analysis shows that the surface microstructure was generated by the combined action of polishing fluid scouring and cavitation jet erosion, and the cavitation jet erosion was stronger than the material removal effect of high-frequency scouring.

Figure 11 shows that the surface roughness at the machining edge was larger than the center area and had more obvious dispersion. The surface microstructure at the edge groove significantly reduced the dispersion and surface roughness. According to the research results of Section 2, it can be known that the pressure fluctuation at the machining edge was smaller than that of the machining center, the blast density of cavitation bubbles was reduced, and a more favorable growth into larger bubbles with greater bursting power occurred. Therefore, the removal uniformity was poor and the surface roughness was higher at the edge. The flow velocity at the groove was large and the scour of the polishing fluid played a certain role in reducing the surface roughness. The roughnesses in the central area and the grooves were not much different, but the area in red was larger in the figure, which indicated better removal uniformity. This was due to the higher density of cavitation bubbles bursting in the central region under high pressure fluctuations, but the bursting energy of individual bubbles was reduced.

Figure 12a shows that the removal depth decreased exponentially and the surface roughness increased linearly with the increase in the gap. When the removal depth was larger, the measurement error caused by the unevenness of the starting surface was larger, and thus, the error bar range was larger. Figure 12b shows that as the gap increased, the red areas representing the microscopic peaks on the surface were lower and the distribution was more dispersed. Based on the simulation results of Section 2, it can be known that the pressure fluctuation amplitude increased as the gap decreased, which caused the cavitation bubbles to be more likely to collapse, which also meant that the burst concentration of the cavitation bubbles was higher and the cavitation bubbles did not grow into strong, large bubbles. A higher bubble burst concentration means more uniform removal. Moreover, when the gap is small, the flow velocity in the gap is larger, which is more conducive to the scouring removal of microstructure peaks.

Figure 13 shows that the removal depth increased linearly with the increase in processing time. The growth rate of the surface roughness became slower with the increase in time, and the larger the gap, the larger the roughness value. It can also be seen from Figure 13b that as the processing time increased, the number and area of the red regions representing the micro-convex peaks decreased (compare time = 300 s and time = 450 s); the blue area representing the lowest point was also decreasing (compare time = 450 s and time = 600 s). This was because the initial surface roughness of the sample used was very low (Sa = 0.5 nm) and the initial surface was not completely removed in the early stage of processing, and thus, the roughness gradually increased. When the newly machined surface formed and the density of effective cavitation bubbles had stabilized, the roughness stabilized.

As can be seen from Figure 14a, as the particle concentration increased, the depth of the removal increased exponentially with a slight decrease in the surface roughness. From the microstructure shown in Figure 14b, the area of the blue region representing the pit was larger at lower concentrations, which indicated that the removal uniformity in the microscopic range was poor. When the concentration increased, the red area representing the convex peak on the microscopic surface increased, which indicated that the scouring removal effect was significantly enhanced. Previous research showed that the presence of particles in liquids can cause bubbles to burst more easily [33]. Therefore, as the particle concentration increased, the burst concentration of cavitation bubbles increased, which contributed to the improvement of removal uniformity.

## 4. Modeling of Material Removal Distributions and Surface Morphology

### 4.1. Material Removal Distributions Model

Based on the experimental results of Section 3.2, each part of the *RF* could be described as follows:(6)$$S_{surface}=H\times S_{Tool}\times t$$
(7)$$H=-k_{1}e^{k_{2}\left(h_{amplitude}-k_{3}h_{gap}\right)}$$
where *H* is the cavitation removal rate; *h_amplitude_* is the amplitude; *h_gap_* is the gap value; *S_tool_* is the shape of the tool’s working surface; and *k_1_*, *k_2_*, and *k_3_* are constants related to the fluid concentration and the optical material.
(8)$$S_{edge}=H\times \frac{R_{edge}*L_{Tool}}{\left|R_{edge}*L_{Tool}\right|}\left(U_{ones}-U_{ones}⋂S_{Tool}\right)\times t$$
(9)$$R_{edge}=-e^{-\frac{x^{2}}{2c^{2}}}$$
where *L_Tool_* is the edge of the tool’s working surface, *R_edge_* is the edge function, * is the convolution operation, and *c* is the width of the edge effect. In this study, the width of the edge effect was *c* = 0.25 mm. *U_ones_* is an all-ones matrix of size equal to *R***L_Tool_*.

In the NUAM processing method, the polishing liquid in the narrow gap mainly moves tangentially on the surface of the workpiece under the drive of ultrasonic vibration, and thus, the material removal distribution realized by fluid scouring can be simplified to
(10)$$S_{scour}=kv^{a}\times t$$
where *k* and *a* are constants and *v* is the normalized near-wall velocity distribution.

By adjusting the relevant parameters in equations 4 to 10 according to the experimental detection results, the geometric model of the tool removal function can be established, as shown in Figure 15.

### 4.2. Evolution of the Surface Micromorphology

The surface of BK7 optical glass processed using NUAM was observed by an atomic force microscope, as shown in Figure 16. According to the three material removal modes, the surface microtopography shown in Figure 16 could be divided into deep pits (mode-A) formed by the impact of cavitation bubble blasting and a large amount of crossing slender scratches (mode-B). It can be seen from the figure that the number of slender scratches in mode-B was significantly higher than the number of pits in mode-A, and the shape of the removed pits in both modes was very irregular. To facilitate the modeling and analysis, this study regarded the erosion pits of mode-A and mode-B formed under the same process parameters as circular pits with Gaussian shape changes and elliptical pits with Gaussian shape changes, respectively. Therefore, the shape of the mode-A pit formed by a single impact can be described as follows:(11)$$RF_{A}=-h_{A}\times e^{-\frac{2r^{2}}{d_{A}^{2}}}$$
where *h_A_* is the depth of the erosion pit and *d_A_* is the diameter of the erosion pit.

The material removal distribution formed by mode-A pits on the workpiece surface per unit time is given by the following formula:(12)$$\sum RF_{A}=RF_{A}*D_{A}$$
where *RF_A_* is the pit formed by a single impact, *D_A_* is the pits distribution, and * is the convolution operation symbol.

The shape of the mode-B pit formed by a single impact can be described as follows:(13)$$RF_{B}=-h_{B}\times e^{-\frac{2x^{2}}{l_{B1}^{2}}-\frac{2y^{2}}{l_{B2}^{2}}}$$
where *h_B_* is the depth of the erosion pit, *l_B1_* is the length of the long side of the erosion pit, and *l_B2_* is the length of the short side of the erosion pit.

The material removal distribution formed by mode-B pits on the workpiece surface per unit time is given by the following formula:(14)$$\sum RF_{B}=\underset{i=0,45,\;90,135}{\sum }\;f\left(\theta_{i}\right)\times RF_{B}*D_{B}^{i}$$
where *RF_B_* is the pit formed by a single impact, *D_B_^i^* is the removal distribution of the angle biased toward *θ_i_*, and *f(θ)* is the rotation matrix. To simplify the analysis, the directions of *RF_B_* are summarized as four directions: 0°, 45°, 90°, and 135°.

The volume of material removed using NUAM processing per unit time can be expressed by the following formula:(15)$$V=\overset{M}{\underset{i=0}{\sum }}RF_{A_{i}}+\overset{N}{\underset{j=0}{\sum }}RF_{B_{j}}+RF_{C}$$
where M is the number of mode-A pits generated per unit time, N is the number of mode-B pits generated per unit time, and *RF_C_* is the material removal caused by polishing liquid scouring. Within the range of process parameters investigated in this study, the material removal rate achieved by abrasive particles under liquid drag is very low compared with mode-A and mode-B, and thus, it is not considered in this section.

According to Figure 16, within the detection range of 2 μm × 2 μm, the diameter *d_A_* of the mode-A pit was about 100–200 nm, the depth *h_A_* was between 14 and 24 nm, and the number M was about 4. The length of the long side *l_B1_* of the mode-B pit was about 100 nm, the length of the short side *l_B2_* was about 10 nm, the depth *h_B_* was between 14 and 24 nm, and the number N was about 400. The evolution process of the surface generated by the combined action of mode-A pits and mode-B pits is shown in Figure 17.

If the generated surface is represented by a matrix *A* of size M × N, the roughness of the generated surface can be calculated according to the following equation:(16)$$S_{a}=\frac{1}{MN}\overset{M}{\underset{i=0}{\sum }}\overset{N}{\underset{j=0}{\sum }}\left|A\left(i,j\right)-mean\left(A\right)\right|$$

Figure 17c shows the surface morphology after processing per unit of time. Since the parameter settings in Table 3 were obtained according to the observation of the microscopic topography shown in Figure 16, the unit time was the processing time of the microscopic topography shown in Figure 16, which was 300 s. By comparing the simulation results shown in Figure 17c with the experimental detection results shown in Figure 16, it can be found that the simulation results were significantly smaller than the measurement results. According to the analysis, this was the difference caused by the fact that the starting surface of the test sample was not an ideal plane. Figure 17f shows that after 100 units of processing time, the surface roughness reached 15.7 nm, which was also in line with the change trend of the processed surface roughness with processing time shown in Figure 13.

Based on the above studies, the evolution trends of machining removal depth and machined surface roughness with machining time are shown in Figure 18.

Figure 18a shows that the surface roughness increased with time and gradually decreased with increasing speed. The surface processed with mode-A and mode-B had a higher roughness than the surfaces processed separately. The surface roughness increase formed by mode-B was very smooth, while the surface roughness increase curve formed using mode-A and mode-A+B had a certain fluctuation, which indicated that the influence of mode-A on the surface roughness was significantly greater than that of mode-B. Therefore, when wanting to obtain a higher machined surface quality, the occurrence of mode-A in machining should be suppressed or avoided, and when wanting to obtain higher machining efficiency, the frequency of occurrence of mode-A should be promoted. How to improve the processing efficiency while satisfying a surface quality requirement, it is necessary to adjust the frequency of occurrence of different modes by optimizing the process parameters.

Figure 18b shows that the removal depth increased linearly with time. The removal depth of mode-A and mode-B processing together equalled the depth when using each processing mode alone. Under the parameter settings listed in Table 3, the material removal rate ratio of mode-A and mode-B was 28:44. The point A, B, and C represent the processing time of different methods for the same removal depth, corresponding to the points with the same name in Figure 18a. The points A, B, and C in Figure 18 show that when the same depth of material was removed, the roughness of the surface formed by mode-A was the largest and the roughness of the surface formed by mode-B was the smallest.

The variation curve of Sa with time *t* in Figure 18a can be represented by fitting the following exponential function:(17)$$S_{a}=a\cdot t^{b}$$
where the larger the *b* is, the greater the rate of increase of the surface roughness with time. The size of the goodness of fit can reflect the fluctuation of the surface roughness curve compared with the function curve.

Since the random distribution function was used in the solution operation, each solution result had a certain fluctuation, but the fluctuation size was not enough to change the influence law, and thus, there is no error range added to the data in the table.

From the data changes in Table 4, the following conclusions can be drawn: Under the same material removal rate, as the proportion of mode-A pits increased, the surface roughness increased. The change in surface roughness under the action of mode-A and mode-B exhibited an obvious exponential change (R^2^ ≈ 1) with the exponential size b ≈ 0.5. The surface roughness change rate caused by mode-A was significantly larger than that of mode-B. The fluctuation of surface roughness was mainly affected by mode-A. As the number of impacts increased, there was no obvious trend in the rate of surface roughness increase. The surface roughness can be reduced by increasing the frequency of mode-A or mode-B per unit removal depth.

## 5. Conclusions

In conclusion, NUAM technology was introduced into the polishing process of optical components, and theoretical and simulation analyses were carried out for the material removal distribution, material removal rate, and surface quality when using precision polishing. The removal function model and the evolution model of the microscopic surface morphology under this processing method were established. Some conclusions can be described as follows:The material removal caused by cavitation bubble explosion was uniformly distributed on the entire working surface and had a 0.25 mm edge influence range. The flow scouring removal was mainly concentrated in the high-velocity flow zone around the machining area.The material removal rate increased exponentially with the decrease in machining gap and the increase in amplitude, and remained constant with machining time. This feature is suitable for deterministic polishing.Under the combined action of cavitation erosion and fluid erosion, the machined surface roughness increased linearly with the increase in amplitude and gap due to the reduced removal uniformity.Increasing the particle concentration significantly improved the material removal rate, and the generated surface exhibited a better removal uniformity and a lower surface roughness.Increasing the blasting density of cavitation bubbles while avoiding near-wall blasting, such as increasing the concentration of abrasive particles, could improve the material removal rate and achieve a higher surface quality.

## Figures and Tables

**Figure 1 micromachines-13-02188-f001:**
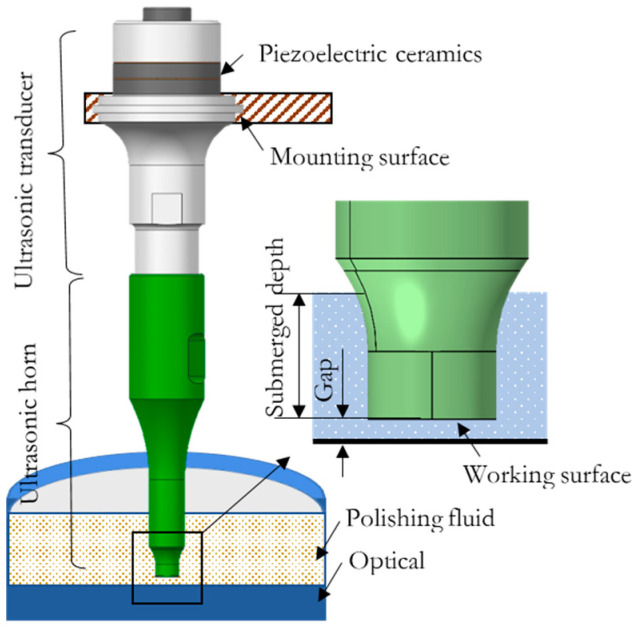
A schematic of non-contact ultrasonic abrasive machining (NUAM).

**Figure 2 micromachines-13-02188-f002:**
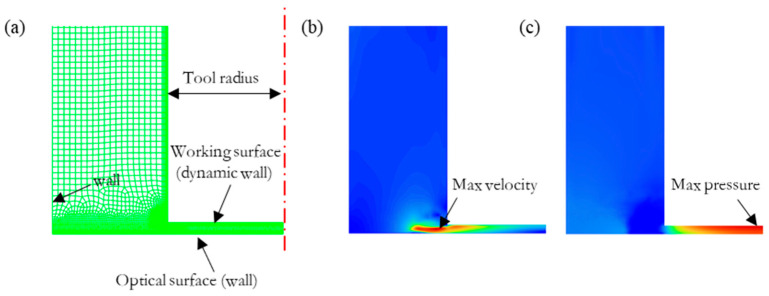
(**a**) FEA mesh model and the (**b**) velocity and (**c**) pressure distributions (gap = 0.25 mm, amplitude = 30 μm).

**Figure 3 micromachines-13-02188-f003:**
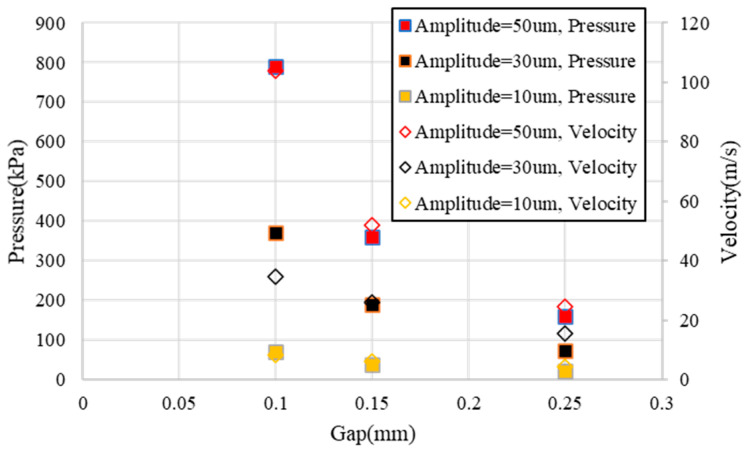
Variation of the maximum speed and pressure with respect to the amplitude and gap.

**Figure 4 micromachines-13-02188-f004:**
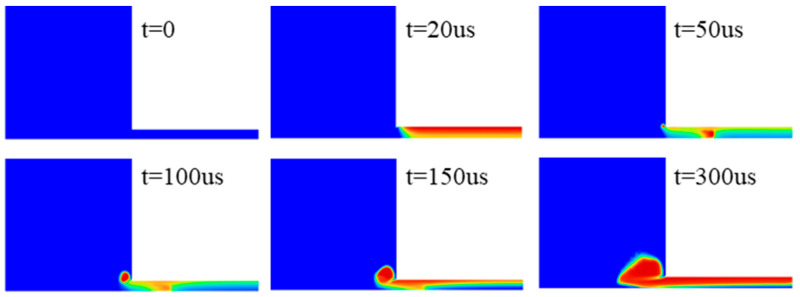
Variation in the vapor fraction over time (gap = 0.25 mm, amplitude = 30 μm).

**Figure 5 micromachines-13-02188-f005:**
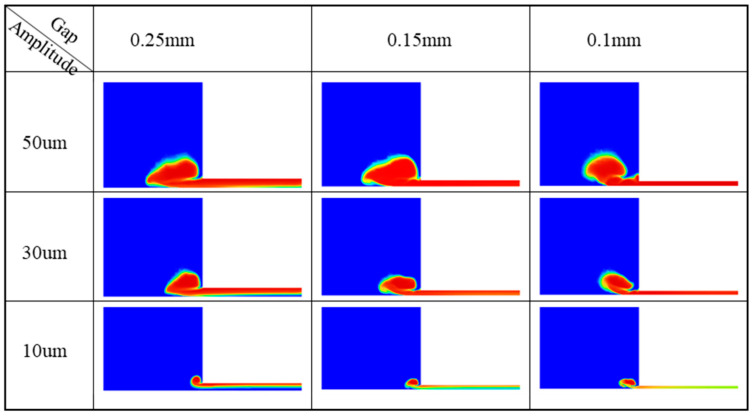
Variation in the vapor fraction with respect to the amplitude and gap (time = 300 μs).

**Figure 6 micromachines-13-02188-f006:**
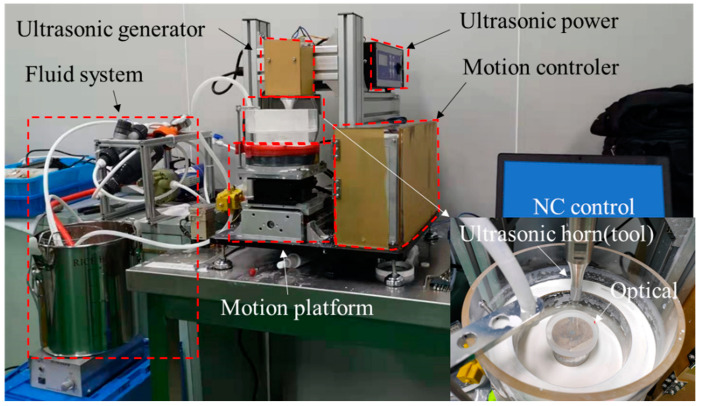
Experimental setup.

**Figure 7 micromachines-13-02188-f007:**
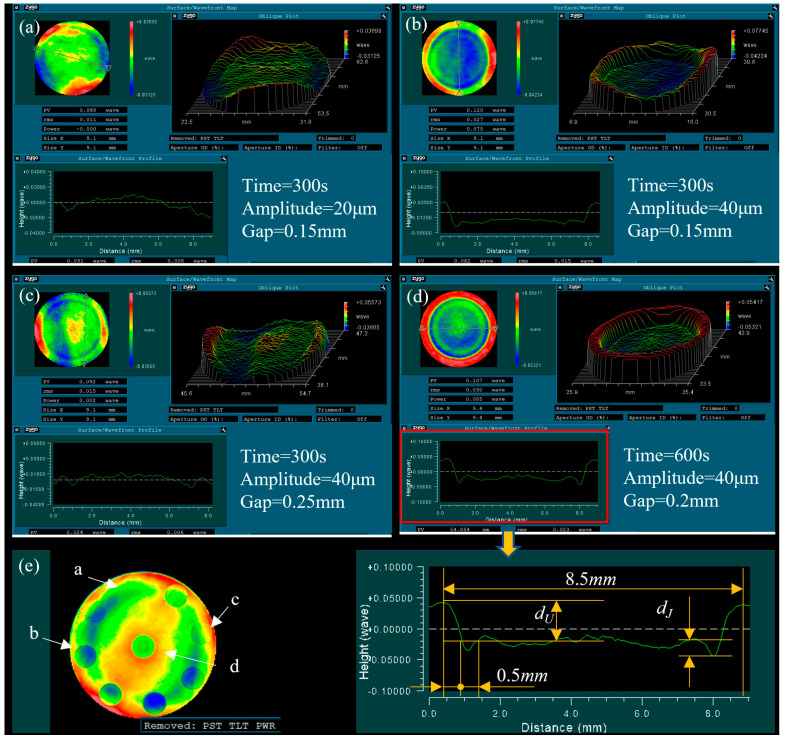
Removal pit profiles after using the ultrasonic tool: (**a**) removal pit under the processing parameters of time = 300 s, amplitude = 20 μm, and gap = 0.15 mm; (**b**) removal pit under the processing parameters of time = 300 s, amplitude = 40 μm, and gap = 0.15 mm; (**c**) removal pit under the processing parameters of time = 300 s, amplitude = 40 μm, and gap = 0.25 mm; (**d**) removal pit under the processing parameters of time = 60 s, amplitude = 4 μm, and gap = 0.2 mm; and (**e**) the lens’s overall surface shape error distribution after the experimental processing.

**Figure 8 micromachines-13-02188-f008:**
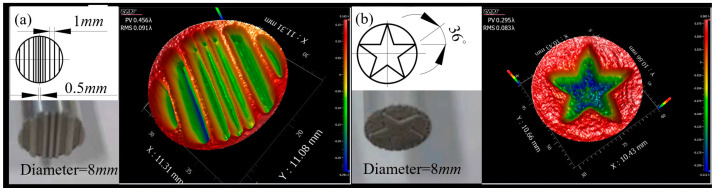
Material removal distribution formed using different geometric structures on the surface of ultrasonic vibration tools: (**a**) rectangle groove and (**b**) five-pointed star (time = 300 s, amplitude = 40 μm, gap = 0.1 mm).

**Figure 9 micromachines-13-02188-f009:**
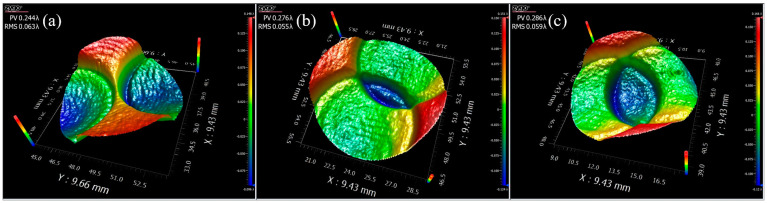
The material removal distribution under different overlap ratios: (**a**) center distance = 8.5 mm, (**b**) center distance = 6.5 mm, and (**c**) center distance = 4.5 mm (time = 300 s, amplitude = 40 μm, gap = 0.1 mm).

**Figure 10 micromachines-13-02188-f010:**
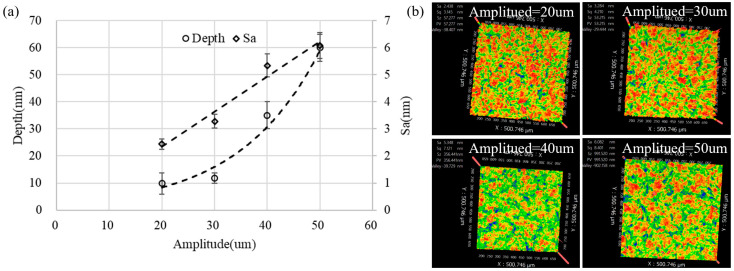
The influence of amplitude on the removal pit: (**a**) the depth and the bottom surface roughness and (**b**) the bottom surface morphology (gap = 0.15 mm, time = 300 s).

**Figure 11 micromachines-13-02188-f011:**
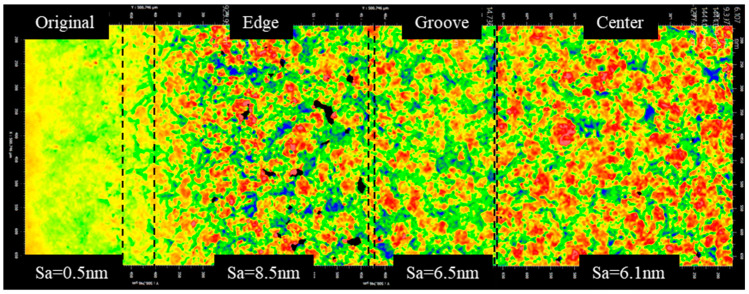
Change in the surface morphology along the radius of the removal pit (gap = 0.15 mm, amplitude = 40 μm, time = 300 s).

**Figure 12 micromachines-13-02188-f012:**
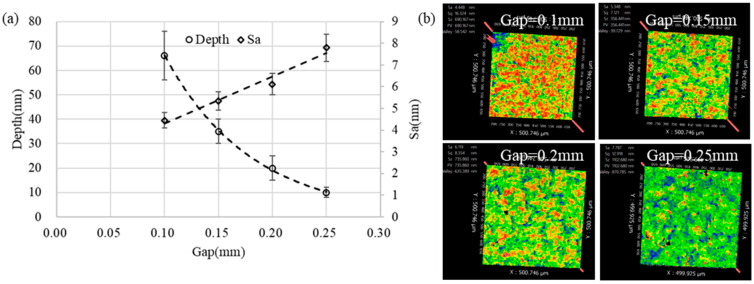
The influence of the gap on the removal pit: (**a**) the depth and the bottom surface roughness and (**b**) the bottom surface morphology (time = 300 s, amplitude = 40 μm).

**Figure 13 micromachines-13-02188-f013:**
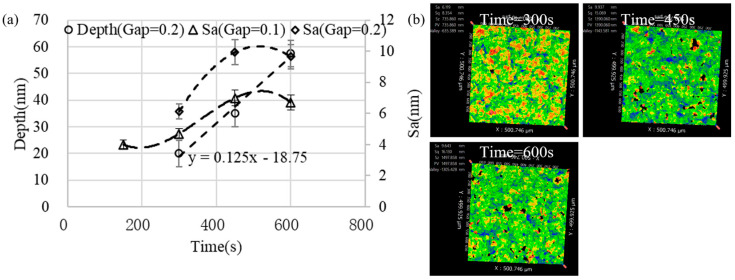
The removal pit changes with processing time: (**a**) the depth and the bottom surface roughness and (**b**) the bottom surface morphology (amplitude = 40 μm, gap = 0.2 mm).

**Figure 14 micromachines-13-02188-f014:**
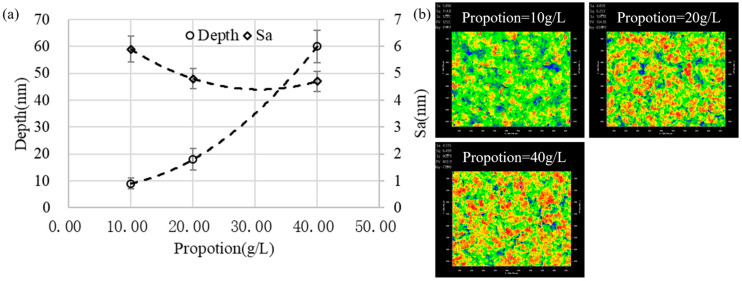
The influence of the fluid proportion on the removal pit: (**a**) the depth and the bottom surface roughness and (**b**) the bottom surface morphology (gap = 0.1 mm, time = 300 s, amplitude = 40 μm).

**Figure 15 micromachines-13-02188-f015:**
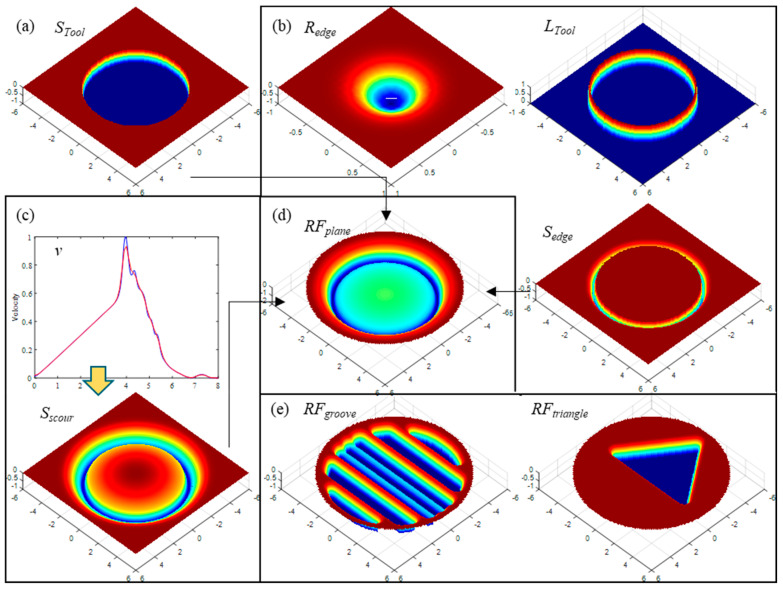
Removal distributions model: (**a**) the shape of the tool’s working surface, (**b**) the edge removal distribution model, (**c**) the fluid scouring removal model, (**d**) the plane tool’s removal function, and (**e**) the tool’s removal function with complex working surfaces.

**Figure 16 micromachines-13-02188-f016:**
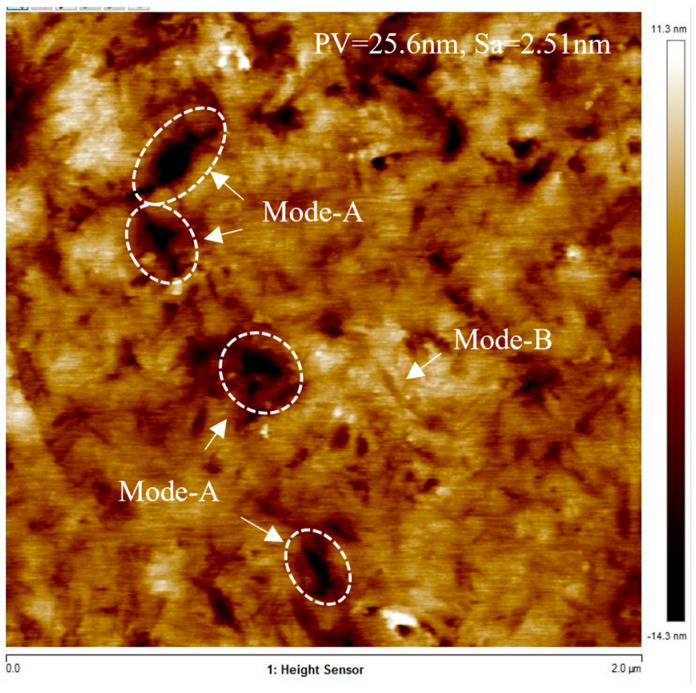
AFM images of the finished surfaces (time = 300 s, gap = 0.15 mm, amplitude = 30 μm).

**Figure 17 micromachines-13-02188-f017:**
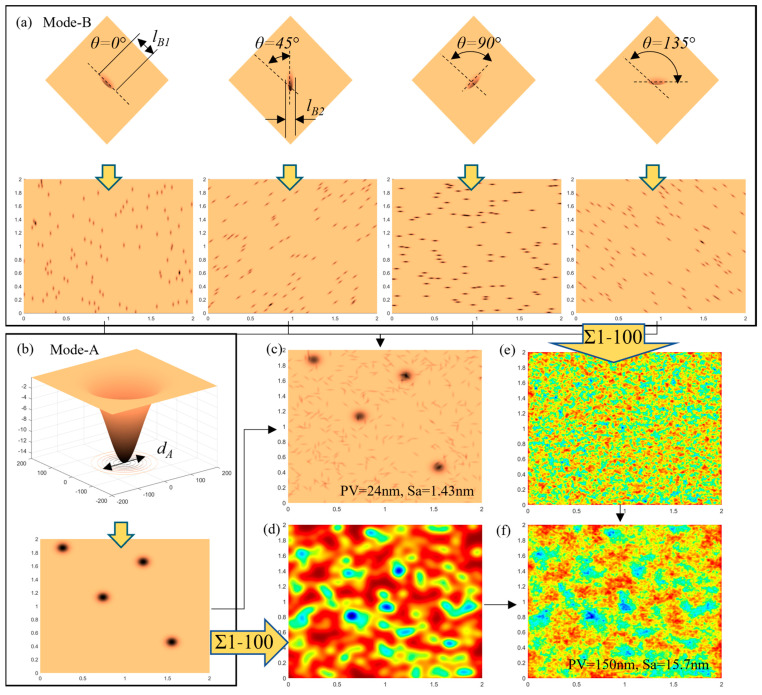
Removal pit shape and formed surface morphology in different modes: (**a**) four directions of mode-B removal pit and a random distribution, (**b**) mode-A removal pit and a random, distribution, (**c**) surface morphology using mode-A and mode-B per unit of time, (**d**) surface morphology using mode-A after 100 units of time, (**e**) surface morphology using mode-B after 100 units of time, and (**f**) surface morphology using mode-A and mode-B after 100 units of time.

**Figure 18 micromachines-13-02188-f018:**
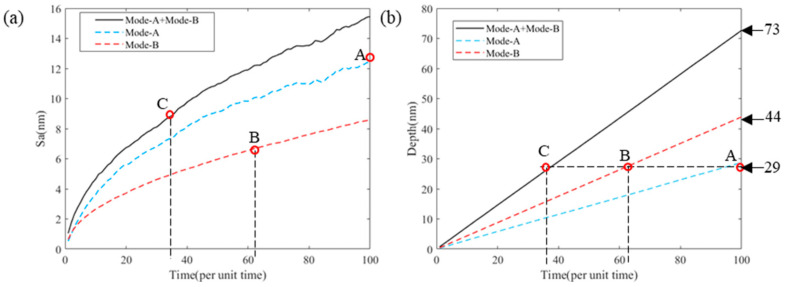
Surface roughness (Sa) and removal depth change with the number of iterations: (**a**) the influence of different modes on the surface roughness and (**b**) the influence of different modes on removal depth.

**Table 1 micromachines-13-02188-t001:** Simulation parameters.

**Amplitude (μm)**	10, 30, 50	**Gap (mm)**	0.1, 0.15, 0.25
**Frequency**	28 kHz	**Fluid**	Water
**Tool radius**	4 mm	**Temperature**	20 °C
**Homogeneous models**	2 Eulerian phases	**Viscous model** [33]	k-omega (SST)
**Cavitation model**	Schnerr–Sauer	**Vaporization pressure**	3540 Pa
**Element size**	Minimum 0.01 mm Maximum 0.2 mm		

**Table 2 micromachines-13-02188-t002:** Experimental conditions.

**Resonance frequency (kHz)**	28
**Amplitude (μm)**	20, 30, 40, 50 (peak to peak)
**Abrasive**	CeO_2_ (1 μm in mean diameter)
**Machining fluid**	Water
	Water mixed with abrasive grains (10 g/L)
	Water mixed with abrasive grains (20 g/L)
	Water mixed with abrasive grains (40 g/L)
**Machining time (s)**	150, 300, 450, 600
**Gap (mm)**	0.1, 0.15, 0.2, 0.25
**Fluid temperature (°C)**	25
**Workpiece**	BK7 flat optical glass
	Surface roughness (Sa): 0.5 nm
	Hardness (HK100): 610 kg/mm^2^
	Dimensions: φ 50 × 10 mm
**Submerged depth (mm)**	20
**Working surface on the horn**	Plane (diameter: 8 mm)
	Rectangle groove (diameter: 10 mm)
	Five-pointed star (diameter: 10 mm)
**Testing equipment**	Zygo GPI for material removal distribution
	Zygo Newview9000 for surface morphology

**Table 3 micromachines-13-02188-t003:** Surface morphology generation simulation parameters.

Mode-A	Diameter	*d_A_* = 150 nm
	Depth	*h_A_* = 19 nm
	Number	*M* = 4
Mode-B	Major axis	*l_B_*_1_ = 100 nm
	Minor aixs	*l_B_*_2_ = 10 nm
	Depth	*h_B_* = 5 nm
	Number	*N* = 400

**Table 4 micromachines-13-02188-t004:** Changes in the surface roughness caused by the proportion of pits in different removal modes.

No.	M	N	Sa (nm)	Depth (nm)	Sa/Depth	a	b	Goodness of Fit (R^2^)
1	3	466	15.16	72.75	0.208	1.287	0.532	0.9991
2	3	0	11.51	21.37	0.54	0.8873	0.563	0.9947
3	0	466	9.35	51.38	0.18	0.8842	0.5089	0.9998
4	4	400	15.69	72.37	0.217	0.133	0.5482	0.9992
5	4	0	13.08	28.45	0.46	1.073	0.5561	0.9966
6	0	400	8.539	43.92	0.19	0.7974	0.5166	0.9998
7	5	334	16.16	72.46	0.223	1.488	0.5183	0.9983
8	5	0	14.23	35.58	0.4	1.236	0.5327	0.9972
9	0	334	7.73	36.89	0.21	0.7609	0.5053	0.9996

## Data Availability

Not applicable.

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
