# Peer review of "Investigation of Material Removal Distributions and Surface Morphology Evolution in Non-Contact Ultrasonic Abrasive Machining (NUAM) of BK7 Optical Glasses"

_micromachines, 2022, doi:10.3390/mi13122188_

Round 1

Reviewer 1 Report

In the paper entitled "Investigation of Material Removal Distributions and Surface Morphology Evolution in Non-Contact Ultrasonic Abrasive Machining (NUAM) of BK7 Optical Glasses", the authors introduced the NUAM technology to process the BK7 optical glass. Here are some questions which may help improve the manuscript as follows:

(1)     Grammar mistakes and nonstandard writings need to be carefully revised. For instance, there should be a space between the number and the unit; “2” in “CeO2” should be subscript. What does “removed PWR” mean?

(2)     Section 2 theoretically analyzed the material removal distribution under NUAM, while Section 3 performed NUAM experiments. However, it isn't easy to find the correlation between Section 2 and Section 3. Please improve it.

(3)     Did the authors derive the equations in the paper? If not, the authors should cite the references in the right places.

(4)     The description of the experimental results in Section 3 lacks an in-depth discussion of the underlying mechanism. For instance, “Figure 10(a) shows that the removal depth increases exponentially, and the surface roughness increases linearly with the increase of amplitude.” Why did the data have such specific relationships? Besides, some conclusions lack direct evidence or references. For instance, “This is due to the higher density of cavitation bubbles bursting in the central region under high pressure fluctuations, but the bursting energy of individual bubbles is reduced.” Please improve them.

(5)     What is the relationship between Section 2 and Section 4? They are both like theoretical analysis of the material removal distribution. Moreover, the authors should compare the experimental results in Section 3 with the theoretical analysis in Section 2 and Section 4.

(6)     It can be seen that the surface quality deteriorated with NUAM. In addition, NUAM has a noticeable edge effect, which may harm the accuracy to shape. In practice, surface quality and shape accuracy are critical to the performance of optical glasses. What are the advantages of NUAM compared with other high-precision polishing techniques, such as CCOS, magnetorheological finishing, and ballonet polishing?

Author Response

some questions which may help improve the manuscript as follows:

  • Grammar mistakes and nonstandard writings need to be carefully revised. For instance, there should be a space between the number and the unit; “2” in “CeO2” should be subscript. What does “removed PWR” mean?

Response 1: Thanks for your advice.

We hired professionals to check and polish the writing of this article.

All formatting errors mentioned above have been carefully corrected and checked.

The "removed PWR" was forgotten to be deleted during the revision process of the manuscript, and it has been deleted.

  • Section 2 theoretically analyzed the material removal distribution under NUAM, while Section 3 performed NUAM experiments. However, it isn't easy to find the correlation between Section 2 and Section 3. Please improve it.

Response 2: Thanks for your advice.

In the section 2, we analyze the material removal distribution caused by cavitation and polishing fluid scouring through theoretical analysis and computer simulation.

The research results of Section 2 are the basis for the analysis of experimental results. To better analyze the experimental results, the research results of section2 are cited in many places of section 3, like P9 261-264, P10 312-314, P11 333-335 and P12 352-353.

  • Did the authors derive the equations in the paper? If not, the authors should cite the references in the right places.

Response 3: Equations 1 to 4 are quoted from other papers. References are indicated in the description above the formula.

Equations 5 to 15 are self-derived.

  • The description of the experimental results in Section 3 lacks an in-depth discussion of the underlying mechanism. For instance, “Figure 10(a) shows that the removal depth increases exponentially, and the surface roughness increases linearly with the increase of amplitude.” Why did the data have such specific relationships? Besides, some conclusions lack direct evidence or references. For instance, “This is due to the higher density of cavitation bubbles bursting in the central region under high pressure fluctuations, but the bursting energy of individual bubbles is reduced.” Please improve them.

Response 4: Thanks for your suggestion.

There is no direct observation technology of the cavitation bubble concentration and cavitation bubble intensity distribution in the processing area, and it can only be analyzed and speculated through experimental results combined with theory and simulation analysis. To enhance the persuasiveness of the analysis in this paper, explanations and references have been added in the manuscript.

“It can be known from the above analysis that as the amplitude increases, the cavitation density and cavitation intensity also increase, which means that more and stronger cavitation bubbles participate in the material removal, increasing the removal rate.”

“According to the research results of section 2, it can be known that the pressure fluctu-ation at the machining edge is smaller than that of the machining center, the blast density of cavitation bubbles is reduced and a more favorable growth into larger bub-bles with greater bursting power.”

“Based on the simulation results of section 2, it can be known that the pressure fluctua-tion amplitude is increased with the gap decreasing, which causes the cavitation bub-bles more likely to collapse, which also means that the burst concentration of cavita-tion bubbles is higher, and the cavitation bubbles do not grow into strong, large bubbles.”

  • What is the relationship between Section 2 and Section 4? They are both like theoretical analysis of the material removal distribution. Moreover, the authors should compare the experimental results in Section 3 with the theoretical analysis in Section 2 and Section 4.

Response 5: Section 2 is based on the cavitation theory and uses the simulation method to analyze the influence of process parameters on the ultrasonic cavitation field and velocity field. This study is to help analyze the experimental results in section 3, to understand the formation mechanism of the material removal distribution.

The geometric model of the material removal distribution in Section 4 is obtained by simplifying the experimental results, so no comparison is carried out. Soon, we will use NUAM technology to carry out surface error correction research. In this study, a detailed analysis will be carried out on the impact of the deviation between the model prediction and the experimental results. In the geometric model of the surface morphology in Section 4, since the material removal in different modes is randomly distributed, it is meaningless to compare the geometric surface profile between the simulation results and the experimental results, but the statistical data of the surface shape can be compared, such as the roughness value and peak-to-valley value within the measurement range.

For comparison, we added surface roughness and peak-to-valley values in Fig. 16, Fig. 17(c) and Fig. 17(f). and gave an explanation under the Fig. 17.

“Figure 17c shows the surface morphology after processing of per unit time. Since the parameter settings in Table 3 are obtained according to the observation of the mi-croscopic topography shown in Figure 16, the unit time is the processing time of the microscopic topography shown in Figure 16, which is 300s. By comparing the simula-tion results shown in Figure 17c with the experimental detection results shown in Fig-ure 16, it can be found that the simulation results are significantly smaller than the measurement results. According to the analysis, this is the difference caused by the fact that the starting surface of the test sample is not an ideal plane. Figure 17f shows that after 100 times of processing unit time, the surface roughness reaches 15.7nm, which is also in line with the change trend of the processed surface roughness with processing time shown in Figure 13.”

(6)     It can be seen that the surface quality deteriorated with NUAM. In addition, NUAM has a noticeable edge effect, which may harm the accuracy to shape. In practice, surface quality and shape accuracy are critical to the performance of optical glasses. What are the advantages of NUAM compared with other high-precision polishing techniques, such as CCOS, magnetorheological finishing, and ballonet polishing?

Response 6: To analyze the removal mechanism and removal characteristics of the material, the experiments of NUAM machining were carried out on the smooth surface after fine polishing, so it showed that the surface quality deteriorated. The NUAM technology is planned to process the surface after grinding, and the surface quality processed by this technology is obviously better than that after grinding.

The edge influence refers to the processing area edge of the tool, not the edge of the workpiece. In computer controlled deterministic polishing, if the material removal function can be kept stable, high precision surface error correction can be performed.

Compared with other polishing methods, the structure of the NUAM device is simple, and the material removal rate can be significantly improved by adjusting the gap or amplitude, so it has greater application potential.

Reviewer 2 Report

This paper investigated the evolution mechanism of material removal distribution in Non-contact ultrasonic abrasive machining process. This paper is well organized, and the proposed method is a potential way to realize the machining of complex structure. Thank you for the opportunity to review this interesting work. Manuscript can be accepted for publication after revision considering the following recommendations. My comments are as follows:

Simulation section:

1.      What is the density of mesh? Does it affect the simulation results?

2.      The axisymmetric two-dimensional model is used in the manuscript. How could the authors ensure that the two-dimensional model has the acceptable accuracy and precision as the three-dimensional model?

3.      The repeated narrative in P4 line 145 & 149, please check it kindly.

4.      P6 line 200: “A ring distribution of vibration velocity was found to match reasonably well the experimental data in the ultrasonic FJP.” Is there any evidence to support this conclusion?

Experimental section:

1.      Why does the cross section outline in Figure 7C not pass the center of the circle?

2.      The article mentions “the ratio of the pit removal depth dJ to the average removal depth dU” many times. Can it be quantified?

3.      In the experimental studies conducted in this paper, is the width of the affected area of RF edge effect obtained in all experiments 0.25mm? If so, give evidence; If not, how to ensure the "deterministic polishing" mentioned in the article?

4.      “Gap=0.2mm” in Fig. 13b can be placed in the figure caption.

5.      In section 4.2, evolution of the surface micromorphology is introduced by deep pits (Mode-A) and crossing slender scratches (Mode-B) in AFM images of the finished surfaces (time=300s gap=0.15mm amplitude=30um). But at the end of this section, why not give a comparison of the evolutionary model with the experimental results?

Some formatting issues:

1.      Microns occur as ‘um’, ‘μm’ and ‘micron’ in the manuscript, please unify the format of units.

2.      ‘Gap=xx’ appears in the text and figure captions without units, please complete.

3.      Many upper and lower scripts in this article indicate irregularities, such as ’CeO2’, ’R2’. Please check and correct them.

Author Response

  1. What is the density of mesh? Does it affect the simulation results?

Response 1: Thanks for your review.

The mesh density is refined at the gap and wall locations. The minimum element size is 0.01mm and the maximum element size is 0.2mm. We have added this parameter to Table 1.

We conducted simulation studies with different element sizes and found that the effect of smaller mesh sizes on the simulation results was not significant.

  1. The axisymmetric two-dimensional model is used in the manuscript. How could the authors ensure that the two-dimensional model has the acceptable accuracy and precision as the three-dimensional model?

Response 2: There are two main reasons why we use a 2D model for simulation analysis. First, the ultrasonic vibration we use in this paper is perpendicular to the surface of the workpiece, and the shape of the tool is also an axisymmetric circular plane, so the excited fluid flow is also Mainly radial movement. Second, the simulation results are only used to analyze the change trend, and there is no quantitative requirement.

  1. The repeated narrative in P4 line 145 & 149, please check it kindly.

Response: Thanks for your checking. The repeated description has already removed.

  1. P6 line 200: “A ring distribution of vibration velocity was found to match reasonably well the experimental data in the ultrasonic FJP.” Is there any evidence to support this conclusion?

 Response 4: This sentence forgetting to delete due to our mistakes in reviewing the manuscript. It has been removed.

Experimental section:

  1. Why does the cross section outline in Figure 7C not pass the center of the circle?

Response 1: Because the removal depth under this parameter is small, the initial surface shape error has a greater impact on the measurement of the removal depth. To measure the removal depth, a relatively flat location was selected for data collection.

The figure below shows a section line through the center. It is difficult to identify the edge position and the removal depth from it.

  1. The article mentions “the ratio of the pit removal depth dJto the average removal depth dU” many times. Can it be quantified?

Response 2: The ratio of the pit removal depth dJ to the average removal depth dU shows an obvious change with the change of process parameters. Due to the impact of the initial surface shape error of the lens used in the experiment, it is difficult to accurately extract the pit removal depth when the removal depth is small. Therefore, this paper does not conduct an in-depth analysis of the ratio.

Thank you very much for your reminder, we will study this ratio in future work.

  1. In the experimental studies conducted in this paper, is the width of the affected area of RF edge effect obtained in all experiments 0.25mm? If so, give evidence; If not, how to ensure the "deterministic polishing" mentioned in the article?

Response 3: Thank you very much for your question.

In the fixed-point machining experiment shown in Figure 7, the edge influence range measured in all experimental results is 0.25mm. When the removal depth is small, the edge is difficult to identify, so there may be some errors. In addition, there are some evidences to prove that, for example: the groove with a width of 0.5mm shown in Figure 8a has a residual shape of a line on the processed surface. The sharp angle of the five-pointed star on the surface of the workpiece shown in Figure 8b is an arc with a radius of 0.25mm on the processing surface. As shown in Figure 9a, the distance between the tool edge is 0.5mm for two times processing, and the removal edge is also just in contact.

  1. “Gap=0.2mm” in Fig. 13b can be placed in the figure caption.

Response 4: Thanks, we have modified it according to your suggestion.

  1. In section 4.2, evolution of the surface micromorphology is introduced by deep pits (Mode-A) and crossing slender scratches (Mode-B) in AFM images of the finished surfaces (time=300s gap=0.15mm amplitude=30um). But at the end of this section, why not give a comparison of the evolutionary model with the experimental results?

 Response 5: In the microscopic surface formation simulation model, the material removal in different modes is randomly distributed, so it is meaningless to compare the geometric surface profile between the simulation results and the experimental results, but the statistical data of the surface shape can be compared, such as the roughness value and peak-to-valley value etc. For comparison, we added surface roughness and peak-to-valley values in Fig. 16, Fig. 17(c) and Fig. 17(f). and gave an explanation under the Fig. 17.

“Figure 17c shows the surface morphology after processing of per unit time. Since the parameter settings in Table 3 are obtained according to the observation of the mi-croscopic topography shown in Figure 16, the unit time is the processing time of the microscopic topography shown in Figure 16, which is 300s. By comparing the simula-tion results shown in Figure 17c with the experimental detection results shown in Fig-ure 16, it can be found that the simulation results are significantly smaller than the measurement results. According to the analysis, this is the difference caused by the fact that the starting surface of the test sample is not an ideal plane. Figure 17f shows that after 100 times of processing unit time, the surface roughness reaches 15.7nm, which is also in line with the change trend of the processed surface roughness with processing time shown in Figure 13.”

Some formatting issues:

  1. Microns occur as ‘um’, ‘μm’ and ‘micron’ in the manuscript, please unify the format of units.

Response 1: Thanks a lot, we double checked and revised the format.

  1. ‘Gap=xx’ appears in the text and figure captions without units, please complete.

Response 2: Thanks, all errors have been checked and corrected.

  1. Many upper and lower scripts in this article indicate irregularities, such as ’CeO2’, ’R2’. Please check and correct them.

Response 3: Thanks, we checked and corrected the formatting errors.

Reviewer 3 Report

1. Please give the full name of VOF.

2. What is α for in Eq(1), and What is V for in Eq(4)?

3. Revise um to μm.

4. Explain briefly why k-omega (SST) viscous model was employed in the simulation model.

5. The number 2 in CeO2 should be in the form of subscript, and 2 in mm2 should be in the form of superscript.

6. Give the subtitles of the images in Figure 7. Fig.7e and Fig.7f should be described in the text.

7. The unit of gap 0.1 is missing in Figure 8. Give the size of the tool used in Figure.

8. g\L should be revised to g/L in Figure 14.

9. Figures are not clear enough.

10. It is better to compare the simulation results in Figure 15 with experimental ones.

Author Response

  1. Please give the full name of VOF.

Response 1: Thanks for your review.

The VOF is the abbreviation of ‘the volume of fluid’, and this explanation was added to the manuscript.

  1. What is α for in Eq(1), and What is V for in Eq(4)?

Response 2: α is the void fraction and it defined as the volume of vapor divided by cell volume.

V is the volume of eroded material by single abrasive.

The above explanations have been added to the manuscript for easy understanding.

  1. Revise um to μm.

Response 3: All the mistakes have already been changed.

  1. Explain briefly why k-omega (SST) viscous model was employed in the simulation model.

Response 4: The k-ω (SST) turbulence model is a two-equation eddy-viscosity model which has become very popular. The use of a k-ω formulation in the inner parts of the boundary layer makes the model directly usable all the way down to the wall through the viscous sub-layer, hence the k-ω (SST) model can be used as a Low-Re turbulence model without any extra damping functions. At the same time, we also refer to the parameter settings of other researchers, like the reference of 33.

[33] Teran, L.A., S. Laín, and S.A. Rodríguez, Synergy effect modelling of cavitation and hard particle erosion: Implementation and validation. Wear, 2021. 478-479.

  1. The number 2 in CeO2 should be in the form of subscript, and 2 in mm2 should be in the form of superscript.

Response: Thanks for your advice. All the mistakes have already changed

  1. Give the subtitles of the images in Figure 7. Fig.7e and Fig.7f should be described in the text.

Response: Thanks for your checking. The subtitles have already added in the title of Figure 7 and give a description under the figure.

  1. The unit of gap 0.1 is missing in Figure 8. Give the size of the tool used in Figure.

Response 7: Thanks for your checking, we have added the unit and diameter value of the tool in figure8 and its title.

  1. g\L should be revised to g/L in Figure 14.

Response 8: Thanks, this mistake has already corrected.

  1. Figures are not clear enough.

Response 9: Thanks for your checking, all the figures have already been replaced.

  1. It is better to compare the simulation results in Figure 15 with experimental ones.

Response 10: Thanks for your suggestion.

The geometric model (Figure 15) of the material removal distribution in Section 4 is obtained by simplifying the experimental results, so no comparison is carried out. Soon, we will use NUAM technology to carry out surface error correction research. In this study, a detailed analysis will be carried out on the impact of the deviation between the model prediction and the experimental results.

Round 2

Reviewer 1 Report

1)    Minor mistakes need to be carefully revised.

2)    The relationship among Section 2, 3, and 4 could be improved.

Author Response

Response 1: Thanks for your checking.

We have carefully checked the manuscript again and corrected any founded errors.

2)    The relationship among Section 2, 3, and 4 could be improved.

Response 2: Thanks for your suggestion.

To improve the relationship between section 2 3 4, the research results of section 2 were cited many times in section 3 to analyze the experimental results, while the theoretical model established in section 4 was completely based on the experimental results in section 3.

A reliable analysis of the material removal distribution and the origin of the surface morphology (section 3) can only be performed with a good understanding of the cavitation density distribution and fluid velocity distribution in the processing area of the NUAM method (section 2). Since the existing theory about cavitation cannot solve quantitative results, the established material removal distribution model and surface microscopic model (section 4) are based on the experimental results (section 3).

To clarify the relationship between each section, explanations are given in many places in the text, such as: P9 260-261, P10 311-313, P11 332-334, P12 351-352, P13 387, P14 408-409, P16 470-479.

Reviewer 2 Report

No comments

Author Response

Thanks for your review.

Best wishes,